# Effects of the Mean Weight of Uniform Litters on Sows and Offspring Performance

**DOI:** 10.3390/ani13193100

**Published:** 2023-10-04

**Authors:** Rui Charneca, Amadeu Freitas, José Nunes, Jean Le Dividich

**Affiliations:** 1MED—Mediterranean Institute for Agriculture Environment and Development & CHANGE—Global Change and Sustainability Institute, Universidade de Évora, Pólo da Mitra, Ap. 94, 7006-554 Évora, Portugal; aagbf@uevora.pt (A.F.); jnunes@uevora.pt (J.N.); 2Independent Researcher, 35590 Saint-Gilles, France; jean.ledividich@club-internet.fr

**Keywords:** litters, piglets, uniformity, colostrum, pre-weaning mortality

## Abstract

**Simple Summary:**

Piglet survival and performance until weaning are key features of sow productivity and, therefore, swine production efficiency. The genetic improvement of the last decades has led to larger litters but also to more heterogenous ones, with an increased number of low birthweight piglets that are more prone to die before weaning due to their lower capacity to ingest colostrum and milk in the competitive environment with their littermates. Some studies show that more uniform litter can be beneficial to piglet survival; however, selection for litter uniformity can lead to smaller litter or birth weight reduction. In this study, we aimed to check the influence of different mean weights of uniform litters on piglet survival and performance. As main conclusions we observed lower pre-weaning mortality in uniform than in heterogenous litters, the colostrum yield (CY) of sows is dependent on the total weight of the litter and litter weight gain in the first day after farrowing is a good marker for CY. The mean weight of piglets of uniform litter influences their colostrum intake and the weaning weight but not their survival. According to the present results, selection for litter uniformity is advisable due to its beneficial effects on sow productivity.

**Abstract:**

This study aimed to determine the effects of uniform litters of different mean birth weights on colostrum production of sows and piglets performance. The study involved 98 multiparous sows from a commercial lean genotype and their piglets. Simultaneous farrowing were supervised and the piglets were divided into experimental litters of 12 piglets each of heterogenous litters (HET, CV = 23.8%, *n* = 20), uniform light litters (ULL, CV = 9.8%, *n* = 27), uniform average litters (UAL, CV = 8.2%, *n* = 23) or uniform heavy litters (UHL, CV = 8.6%, *n* = 28) piglets and allowed to suckle. Piglets were re-weighed at 24 h and 21 d of life and deaths registered. Colostrum intake (CI) of the piglets and sow’s colostrum yield (CY) was estimated using two prediction equations. Significant differences (*p* < 0.001) were observed in the CY of sows being higher in UHL, lower in ULL and intermediary in HET and UAL litters. CY was positively related to litter total weight at birth and litter weight gain in the first 24 h (*p* < 0.001). The CI differ between litter type being higher in UHL litters and lower in ULL litters. The coefficient of variation of CI in HET litters was higher than in uniform litters, regardless of their type. The mortality rate of piglets until 21 d was globally 9.6% and it was significantly higher in HET than in UAL (*p* = 0.033) and tended to be higher than in UHL litters (*p* = 0.052). No differences in piglet survival were observed between uniform litters. Results show the beneficial effect of uniformity in piglet survival and that the mean weight of uniform litter influences colostrum intake and piglet performance.

## 1. Introduction

The number of weaned piglets per year is a key indicator in swine breeding, determining its profitability. During the last decades, genetic selection has led to a significant litter size increase [1]. However, larger litters are associated with higher piglet pre-weaning mortality [2,3,4] which is a major problem in commercial pig production, both from an economic and welfare point of view [5]. With increasing litter size, average birth weight decreases, the within-litter variation in birth weight increases [4,6] and the number of low birthweight piglets per litter also increases [6,7,8]. However, high piglet birth weights and litter uniformity are important for piglet survival and performance [5].

The within-litter coefficient of variation (CV) of a piglet’s birth weight ranges usually from 18 to 26% [8,9,10,11] and, as above mentioned, is positively correlated with pre-weaning mortality but also with variation in weaning weight [2,12].

High within-litter variation in birth weight is associated with a greater proportion of low birthweight piglets in the litter [3,8]. These low birthweight piglets present more difficulties in coping with the first extrauterine phase challenges because they have less glycogen reserves at birth [13,14] and they also have a larger specific surface, making them more susceptible to hypothermia and hypoglycemia during the first 24 h of postnatal life [7].

Colostrum intake is a key factor for adequate nutritional and/or immunological status [15,16,17]. When compared to their heavier littermates, the lighter piglets are at a disadvantage with regard to access to the mammary glands. Therefore, they consume less colostrum [9,18,19], leading to higher pre-weaning mortality rates [10,19,20,21].

Previous studies from our team [22] have shown that uniform litters set before first suckling presented much lower mortality rates during the nursing period than heterogenous litters, mainly due to a more homogenous colostrum intake. In the same work, we proposed the inclusion of litter homogeneity (in weight) in the selection programs in order to improve piglet survival. However, the selection for more uniform litters can have negative impacts both on the litter size and/or on the mean weight of the piglets [23,24]. In the present study, with preliminary results presented at a conference [25], we aimed to assess the impacts on sow and piglet performance (survival and growth) of different mean within-litter weights.

## 2. Materials and Methods

This study was carried out in accordance with the regulations and ethical guidelines set by the Portuguese Animal Nutrition and Welfare Commission (DGAV—Directorate-General for Food and Veterinary, Lisbon, Portugal) following the 2010/63/EU Directive.

### 2.1. Farm and Animals

The experiment was carried out in a private intensive pig farm located close to Évora (Portugal), with a mean herd of 1000 Large White and Landrace sows (Topigs 20). Multiparous sows were artificially inseminated with Piétrain semen (Top Pi) in a 3-week batch system (130–150 sows per batch). Farrowing took place in the 20 farrowing rooms of the farm, and piglets were weaned on average at 28 days of age. Sow gestations were carried out in groups and 5–7 d prior to the expected farrowing date they were moved to the farrowing rooms. The farrowing crates had a PVC slatted floor (under the sow and piglets) and the nest area was heated by a 175 W infrared lamp. They were also equipped with a sow feeder and low-pressure nipple drinkers (for sow and piglets) with a continuous supply of water.

### 2.2. Management of Sows and Piglets

During gestation, sows were fed 3 kg/day of a standard gestation diet (8.91 MJ NE/kg, 15.1% crude protein, 0.8% lysine) until about day 75 and 3.3 kg until farrowing room entry. Thereafter, feed allowance was gradually reduced until no feeding on farrowing day. After farrowing, sows were fed with 2.2 kg/day of a standard lactation diet (9.62 MJ NE/kg, 16% crude protein and 0.9% lysine). The feed allowance was increased by 1.2 kg each 3 days of lactation, to a maximum of about 7 kg from day 12 of lactation until weaning. According to the standard procedures of the farm, on the 2nd (±1) day after birth, piglets were tail-docked, teeth were ground and piglets were injected with 2 mL of Ferrovet (200 mg of iron dextran + 30 µg of vitamin B12). A solid pre-starter diet (9.86 MJ NE/kg, 17.5% crude protein and 1.32% lysine) was provided to the piglets from 7 days old until weaning. Farrowings were usually monitored by farm workers and oxytocin was administrated via intramuscular injection when the birth interval between piglets exceeded ~1 h.

### 2.3. Experimental Procedures

In this trial, farrowings were not induced and were supervised by the research team. No gilts were used in this trial because a different farming protocol is applied in this case, requiring the use of all functional teats in their first lactation and we would be unable to specify the initial litter size. A total of 98 experimental litters were used from 12 farrowing batches.

At birth, all live-born piglets were roughly dried, weighed to the nearest 1 g using an electronic balance (Kern FTB 15 K 0.5 L) equipped with an integration system, identified (ear tag), and their birth time and sex were registered. Stillborn sex and weight and mummified piglet number were recorded for reproductive data collection. All live-born piglets were then put inside a PVC box, under the infrared lamp and inside the nest area of the farrowing crate, in order to provide an environment close to thermoneutrality. After farrowing completion and depending on the measured weights, the piglets were then divided into experimental litters of 12 piglets each of heterogenous litters (HET, CV = 23.8%, *n* = 20) used as control, uniform light litters (ULL, CV = 9.8%, *n* = 27), uniform average litters (UAL, CV = 8.2%, *n* = 23) or uniform heavy litters (UHL, CV = 8.6%, *n* = 28) piglets and allowed to suckle. The experimental litters’ set-up aimed to have a piglet within-litter CV of heterogenous litters above 20% and equal or lower than 10% on uniform litters and the number of 12 piglets per sow was set in order to ensure that in each litter all piglets had access to functional teats. All supernumerary piglets were adopted by sows not used in the study. The choice of sows was conditioned in order to obtain similar average parity. Piglets with a birth weight lower than 700 g were not used because they are often considered a runt and frequently euthanized. Piglets were then re-weighed (to obtain the initial weight before suckling, BW0) and allowed to freely suckle from the sows (time 0). If the farrowing was too prolonged and there were no sufficient born piglets to set up the experimental litters, the sow was not considered for the study. All experimental piglets were individually reweighed 24 h after time 0 for colostrum intake (CI) estimation. Piglets that died after the experimental litter was allowed to suckle were weighed as soon as they were found dead, with the time interval between death and weighing ranging from a few minutes to about 18 h. These piglets were not necropsied. Piglets were reweighed at the end of the study when they had reached 21 d of age.

### 2.4. Calculations and Analyses

The farrowing duration was considered as the lapse of time between the birth of the first and the last piglet. Individual colostrum intake (CI) of the newborn piglets was estimated from piglet weight variation between birth and 24 h and using the prediction equation of Devillers et al. (2004) [26] and the prediction equation of Theil et al. (2014) [27]. Colostrum yield (CY) during the first 24 h after farrowing was calculated by adding the colostrum intakes for each piglet of the litter using each one of the equations.

### 2.5. Statistical Analyses

Data were analyzed using IBM SPSS Statistics software (version 27, 2020). For an overview of reproductive traits and original litter traits, descriptive statistics were performed. The experimental litters were compared using the general linear model (GLM) procedure with the one-way analysis of variance (ANOVA) using litter type (HET, ULL, UAL, and UHL) as a fixed effect. Batch effect was tested in the first analysis but removed from the model as it was not significant for any of the analyzed traits

After litters were created, there were 50% cross-fostered piglets and 50% resident piglets (considering all litters). Although cross-fostering was made before any suckling, a comparison between cross-fostered and not cross-fostered (resident) piglets was made on their birth and 24 h weight and growth performance. The general linear model (GLM) procedure with the one-way analysis of variance (ANOVA) was used, having piglet type (cross-forest vs. resident) as a fixed factor within each litter type.

In order to compare the maximal and minimal CI between heterogenous and uniform litters, the 2 piglets with the lowest and highest CI were selected in each litter. Two types of piglets were then considered: piglets from heterogenous litters and piglets from uniform litters. As they were very unbalanced in number (40 and 156, respectively) the no-parametric Mann–Whitney U test was used to compare piglet type.

Regression analysis was made to determine the relationship between CY and initial litter weight and a correlation procedure was used between CY and litter weight gain in the first 24 h.

Comparisons of piglet mortality rates were assessed using Chi-squared tests. Unless otherwise mentioned, all values are mean ± standard error of the mean (SEM). Differences were considered significant when *p* < 0.05 and *p*-values between 0.05 and 0.10 were considered trends.

## 3. Results

### 3.1. Original Litter

The reproductive and productive traits of sows and piglets in the original litter are presented in Table 1.

### 3.2. Experimental Litters

#### 3.2.1. Parity

On average, sows of the three experimental groups had a similar parity, 3.8 ± 0.3, 3.9 ± 0.3, 4.5 ± 0.3 and 3.7 ± 0.3, respectively, for HET, ULL, UAL and UHL litters (*p* = 0.272).

#### 3.2.2. Characteristics and Performance of the Experimental Litters

The initial traits, litter weight gain and colostrum yield and intake in the first 24 h are presented in Table 2.

Due to the experimental procedures, the average weight of piglets at cross-fostering and experimental within-litter CV were different between groups. The mean weight of piglets was higher in UHL litters, lower in ULL litters and similar between UAL and HET litters, as litter size was equal for all litters, litter total weights are in accordance with the above-mentioned differences. The within-litter CV was low and similar between uniform litters while heterogenous litters had a significantly higher CV. The litter weight gain in the first 24 h was significantly lower in ULL litters when compared to the others that were similar between them. The colostrum yield (CY) was different between groups considering both estimation equations used. Considering Devilers’s equation [26] mean CY was 4578 ± 829 g (mean ± SD), while using Theil’s equation [27] the correspondent value was 6352 ± 1096 g (mean ± SD). Even if the absolute values of CY obtained by the two equations are diverse, the differences between litter types follow the same trends in both cases with the highest value observed in UHL, the lowest value in ULL and HET and UAL litters presenting intermediary values. The CY was positively and significantly related to the initial litter weight (LW0). Using Devillers’s equation [26]: CY = 1860 (± 426) + 0.164 (± 0.025) × LW0 (R^2^ = 0.304, *p* < 0.001); using Theil’s equation [27]: CY = 1372 (± 436) + 0.301 (± 0.026) × LW0 (R^2^ = 0.583, *p* < 0.001). Litter weight gain in the first 24 h was positively significantly correlated with CY both considering Devillers’s equation [26] (0.932, *p* < 0.001) or Thiel’s equation [27] (0.827, *p* < 0.001).

The average colostrum intake (CI) was 396 ± 73 g using Devilers’s equation [26] and 550 ± 94 g using Thiel’s equation [27]. The mean CI differ between litter type. In general, it was higher in UHL and lower in ULL litters. Considering the bodyweight of piglets, CI per kg of birthweight differences were only observed when using Devillers’s equation [26] with higher relative CI in HLL when compared to UHL litters.

The within-litter variability of CI assessed by the CV showed differences between groups that almost reached statistical significance when using Devillers’s equation [26] and reached statistical significance when using Thiel’s equation [27]. In both cases, the CV of CI in HET litters was higher than in uniform litters, regardless of their type. The piglet’s weight gain from 0 h to 24 h was positively and significantly correlated with CI, 0.923 (*p* < 0.001) and 0.851 (*p* < 0.001) using Devillers’s [26] or Theil’s equations [27], respectively.

Cross-fostering was made before colostrum intake in order to obtain the experimental litter. Therefore, in each litter, there were resident piglets (RP, that remained with their natural mother) and cross-fostered piglets (CFP, that were from another sow). Globally, 50% of the experimental piglets were cross-fostered. According to litter type, the percentage of cross-fostered piglets was 46% in HET, 59% in ULL, 54% in UAL and 41% in UHL.

Overall, and compared to cross-fostered piglets, the resident piglets were heavier at time 0 (*p* = 0.031), and at 24 h (*p* = 0.048); however, there were no significant differences in weight gain in the first 24 h (*p* = 0.969) nor on colostrum intake (*p* = 0.491 and 0.225 using Devillers’s [26] and Thiel’s equations [27], respectively). They also present similar weight at 21 d (*p* = 0.256) and mortality rate between time 0 and 21 d of age (*p* = 0.283). Table 3 presents the piglets’ characteristics and performance according to their litter type and nursing sow.

Considering litter type, few significant differences were found between resident and cross-fostered piglets (Table 3). Within each litter type, they had similar weights at birth and 24 h of age. Cross-fostered piglets in ULL litters presented better performance both during the first day of life and during the studied nursing period. They also presented higher colostrum intake and, consequently, higher weight gain in the first 24 h; they were also heavier at 21 d of life than their resident littermates. The higher weight at 21 d of CFP was also observed in UHL litters. The dead piglets were almost equally distributed between resident and adopted piglets in each litter type (Table 3).

At 21 d of lactation and considering all litters, there were 10.9 ± 0.9 (mean ± SD) alive piglets with 6.2 ± 0.9 kg (mean ± SD). The global mortality rate during the 21 d of lactation was 9.6%. The litter characteristics at 21 days, piglet’s performance and mortality rate during the nursing period according to litter type are presented in Table 4.

The mean weight at 21 d of piglets from ULL litters was significantly lower than the mean weight of piglets from all the other litter types. Consequently, ULL litters were also significantly lighter at this age than the others (with the exception of HET litters). The within-litter CV was more than 20% higher in HET litters when compared to all uniform-type litters; however, this difference did not attain statistical significance.

Comparing the mortality rates, HET litters had significantly higher mortality than UAL (*p* = 0.033), and tended to have higher mortality than UHL (*p* = 0.052) but presented no significant differences to ULL litters (*p* = 0.157). Within the uniform litters, there were no significant differences or tendencies regarding this trait (*p* > 0.10).

In the first 24 h after birth, 3.6% of the piglets died, representing 37.2% of the total mortality until d21. It was not possible to estimate the CI of these piglets, but half of them (21 out of 42) lost weight between suckling start and dead body weighing. The mortality until 4 d after birth represents 67.3% of the total mortality. Almost 3/4 (72.4%) of the piglets that died until the fourth day after birth were lighter than the mean weight of their litter. The piglets that died between d1 and d4 (*n* = 34) had a colostrum intake of 173 ± 123 g (mean ± SD) according to Devillers’s equation [26] or 296 ± 139 g (mean ± SD) according to Theil’s equation [27]. The respective values for all the dead piglets (from d1 until d21) were 252 ± 141 g (mean ± SD) and 386 ± 159 g (mean ± SD).

In HET, ULL and UAL litters, most (from 57% up to 82%) of the piglets lost in the first 24 h were lighter than the mean weight of their litters, while in UHL litters that only occurred in 27% of the cases.

## 4. Discussion

This study was made in sequence with another made by our team in the same farm and conditions [22], where significant differences were observed in the survival rate of piglets on uniform litters when compared to heterogenous litters. As in the present study, in the previous one, cross-fostering was made before piglets were allowed to suckle in an attempt to check the effects of litter type if litters (piglets) were already delivered uniform or heterogenous. However, the selection for more uniform litters can have negative impacts both on the litter size and/or on the mean weight of the piglets [23,24]. Increased litter size has been a goal of genetic selection progress in the last decades and, according to recent evaluations, it can continue to be a strategic goal in pig genetic improvement programs because the negative effects on preweaning mortality of piglets can be controlled [1]. The possible negative impacts on the mean weight of the piglets can be also problematic because lighter piglets have low survival rates [28,29,30] and, due to the positive relationship between birth weight and weaning weight [30,31,32], piglets could be weaned with less body weight, resulting in augmented difficulties to cope the weaning transaction challenges [33,34].

In our previous study [22], the uniform and heterogenous litters had, at birth, an average mean weight; in the present study, we aimed to check if there were differences in survival rate and/or performance (growth rate) when the mean weight of uniform litters was higher or lower than the average. We also included heterogenous modified litters (also with cross-foresting to be in a similar situation to the uniform ones) as control.

As determined by the experimental protocol, all the uniform litter types had a coefficient of variation (CV) of the piglet’s weight lower than 10%. The heterogenous litters had a corresponding value of about 23%, which is in the value range for non-intervened (natural) litters as reported in several studies, ranging from 18 to 26% [8,9,10,11]. The initial mean weight of the piglets was also different because of the experimental protocol (light and heavy uniform litters were set), but it was similar between heterogenous and uniform average (UAL) litters. The same protocol is also responsible for the correspondent differences in the total weight of the litter at the beginning of the suckling period.

Performance in the first 24 h, which was assessed by the litter weight gain, colostrum yield by the sow and colostrum intake by the piglets, was different between litter types. Litter weight gain in the first 24 h was significantly lower in ULL litters when compared to all other litter types. Litter weight was positive and significantly related to CY, which was estimated by adding the individual piglet’s colostrum intakes. This significant positive relationship was also found in some studies [35,36,37] but was not observed as being significant by other researchers [38,39]. However, colostrum yield is the result of the colostrum production by the sow and the capacity of its extraction by the piglets and heavier piglets are usually more active and have a higher capacity to extract the mammary gland secretions (colostrum and milk) [19,35,40] which can at least partially explain our findings of higher CY in litters with higher total weight.

According to the observed significant relation between litter weight gain in the first 24 h and CY, the first one can be considered a good marker for sow’s colostrum production. Similar results were found by our team in a previous trial [22].

The global mean CY values in the present study were in the range of those reported by Hasan et al. [41], observed in several farms and genotypes (including Topigs 20 sows) and using Devillers’s equation [26], and those reported by Feyera et al. [42], using Theil’s equation [27].

Colostrum intake means (including all litter types) were similar to the observed by our team on the same farm and with the same genetics [22] and using Devillers’s equation [26]. The mean CI values estimated with Theil’s equation [27] were higher than those recently reported by others [32,43] but besides the differences in piglets’ genetics, the litter size in the cited studies was 15 piglets per litter. With the absence of a positive and significant relation between litter size and CY [35,39], the CI decreases per each additional piglet as observed by [19,35].

At this point, it is worthwhile highlighting that, despite the expected differences in absolute values both for piglet’s colostrum intake and, consequently, for sow’s colostrum yield, when using the two available prediction equations for colostrum intake, the differences between litter types and the relationships between traits were almost identical when using each one of the equations; therefore, the remaining results will be mainly discussed without considering the CI estimation method.

Colostrum intake was lower in ULL litters; however, when considering the intake by kg of birth weight (relative CI), the mean values observed in ULL litters were similar or even higher than those observed in the other litter types. Despite those differences, all mean values for CI were higher than the 200 g per piglet during the first 24 h, a value considered by Quesnel et al. [16] as the minimum consumption to significantly reduce the risk of mortality before weaning, provide passive immunity and allow a slight weight gain or even the 250 g of CI in the same period to achieve all the above and also good health and pre-and post-weaning growth. These reference values were based on estimated CI with Devillers’s equation [26] because Theil’s equation [27] was not yet published. However, the CV of the colostrum intake was higher (about 7 to 10% higher) in HET litters when compared to all uniform-type litters, which means that in HET litters the CI was more variable. In fact, when comparing the CI of the 2 piglets with the lowest CI in each litter, the mean CI in HET litters was significantly lower than the observed in uniform litters (all types together) using either prediction equation. Considering the highest within-litter CI intakes, the differences did not attain statistical significance using Devillers’s equation [26] (*p* = 0.142) but it was significantly higher in piglets from heterogenous litters than from uniform litters (724 ± 18 g vs. 684 ± 9 g, *p* = 0.039) using the values obtained with Theil’s equation [27]. In other words, in HET litters the piglets with low CI consumed much less colostrum than the correspondent piglets in uniform litters which means that these piglets were in a more disadvantageous situation in HET litters. On the contrary, the piglets with high CI are in an advantageous situation in this type of HET litters explaining their higher CI when compared to the correspondent piglets in uniform litters. Similar results were found by our team in a previous trial [22].

This higher variability in CI can explain, at least partially, the observed differences in the mortality rate, higher in HET litters when compared to uniform litter types. Although the global mean colostrum ingestion of piglets with lower within-litter CI was above 200 g (219 g using Devillers’s equation, [26]), about 25% of them had a CI below that referential value. More in detail, the proportion of <200 g CI piglets in HET litters (37.5%) tends (*p* = 0.081) to be higher than the correspondent value in uniform litters (22.4%), which contributes to understanding the mortality rate results.

Checking possible effects of the cross-fostering procedures on the piglet’s survival and/or performance, there were only differences in ULL litters, where cross-fostered piglets had higher CI than resident piglets and, consequently, higher weight gain in the first 24 h. These differences have no apparent explanation and did not lead to significant differences both in mortality rate or body weight at 21 d between piglet types. In UH litters, cross-fostered piglets presented higher body weight at 21 d than resident ones. Again, here we do not find possible explanations for that difference. Taking into account all the results of this comparison, it seems that cross-fostering does not have important positive or negative effects on piglets’ survival or performance.

After a suckling period of 21 d, the piglets from ULL litters were significantly lighter than the piglets from the other litter types, leading to a lower litter total weight. Litter weight positively influences milk production by the sow [40], explaining these results. At 21 days, the within-litter weight CV remained higher in HET litters when compared to uniform litters; however, the differences were no longer significant. The observed slight decrease in CV in HET litters between the beginning of the experiment and day 21 can be related to the death of the lighter piglets of these litters. In fact, 85% of the dead piglets in HET were lighter than the mean weight of HET litters and almost 50% of them weighed less than 1110 g, the threshold proposed by Feldpausch et al. [44] for identifying piglets at risk for preweaning mortality. We also observed a great increase in CV in all uniform litters in the same period. This CV increase in uniform litters (all types) can be explained by differences between individual glands milk production and/or differences in the stimulation efforts of the individual piglets, as previously observed by [12,45]. Milligan et al. [45] also observed that piglet weight variation almost doubled over 21 d in uniform litters and it decreased in heterogenous litters, although it remained higher in the latter.

The global mortality rate until d21 was relatively low (below 10%) and its chronology was similar to the observed in several studies [18,46,47] with an important percentage occurring on the first day (more than 1/3 of the total mortality) and 2/3 of the total mortality occurring until d4. As also observed by others [28,29,32], the main influencing factors for piglet death seem to be its piglet weight, its relative weight (in relation to litter) and the CI in the first 24 h.

Globally, litter uniformization before suckling leads to better results in terms of pre-weaning piglet survival, when compared to heterogenous litters. The mean weight of the uniform litters has no impact on survival rates but low mean weight uniform litters have lower performance than uniform litters of average or heavy mean weights. However, in addition to practical farm management difficulties or impossibilities to homogenize litters before suckling, it is important to mention that cross-fostering as it was made in the present study may have consequences for the piglets because maternal cells (as lymphocytes and epithelial cells) can only be transferred to the piglets if they ingest colostrum from their biological mothers [48]. These cells participate in the antigen-specific response in piglets; therefore, cross-fostering practiced before the first suckling may negatively affect the transfer of cellular immunity and the health of the progeny. Based on the previously discussed constraints and risks of cross-fostering at birth time, the inclusion of litter uniformity in the genetic selection programs may be the best tool to have more homogenous litters leading to a higher preweaning survival rate, thus increasing productivity.

## 5. Conclusions

Previous studies from our team [22] had shown the beneficial effects of having uniform litters at suckling start on pre-weaning piglet survival and litter uniformity at weaning. However, selection for litter uniformity may have detrimental effects on litter size, and/or piglet birth weight. In this study, we aimed to check the possible effects of having uniform litters of different mean weights on the performance and mortality rate during the nursing period. The main results of the present study confirm the beneficial effects of litter uniformity on piglet survival and therefore that litter uniformity should be considered in the breeding programs but also shows that the mean weight of uniform litter influences colostrum intake and piglet performance.

## Figures and Tables

**Table 1 animals-13-03100-t001:** Reproductive and productive traits of sows and piglets of the original litter.

	Minimum–Maximum	Mean	SD
Parity	2–8	3.9	1.5
Farrowing duration (min)	102–451	236	86
Total born	6–21	14.5	3
Born alive	5–18	13.5	2.8
Stillborn	0–6	0.8	1.2
Mummified	0–3	0.2	0.5
Mean birth weight (g) *	940–2193	1395	229
Intra-litter CV (%) *	3.4–36.4	19.7	7.1

* only born alive piglets.

**Table 2 animals-13-03100-t002:** Experimental sows and litter characteristics, colostrum yields and colostrum intakes and coefficient of variation (CV) (means ± SEM).

	Heterogenous (HET)	Uniform Light (ULL)	Uniform Average (UAL)	Uniform Heavy (UHL)	*p*-Value
*n*	20	27	23	28	
Intra-litter CV (%)	23.8 ± 0.6 ^a^	9.5 ± 0.6 ^b^	8.5 ± 0.6 ^b^	8.6 ± 0.5 ^b^	<0.001
Mean piglet weight at time 0 (BW0, g)	1331 ± 25 ^a^	1116 ± 22 ^b^	1397 ± 23 ^a^	1654 ± 21 ^c^	<0.001
Litter weight 0 h (LW0, kg)	16.0 ± 300 ^a^	13.4 ± 258 ^b^	16.8 ± 279 ^a^	19.8 ± 253 ^c^	<0.001
Litter weight gain 0–24 h (LWG, kg)	1.92 ± 0.12 ^a^	1.52 ± 0.10 ^b^	2.03 ± 0.11 ^a^	2.12 ± 0.10 ^a^	<0.001
Colostrum yield (CY, kg) *	4.5 ± 0.2 ^b^	3.9 ± 0.1 ^a^	4.8 ± 0.1 ^bc^	5.1 ± 0.1 ^c^	<0.001
Colostrum intake (CI, g) *	386 ± 14 ^b^	339 ± 12 ^a^	418 ± 13 ^bc^	441 ± 12 ^c^	<0.001
Relative CI (CI/kg BW) *	289 ± 10 ^ab^	308 ± 8 ^a^	300 ± 9 ^ab^	269 ± 8 ^b^	0.017
Colostrum intake CV (%) *	30.4 ± 2.0	23.5 ± 1.8	23.9 ± 1.9	24.3 ± 1.7	0.051
Colostrum yield (CY, kg) **	6.2 ± 0.2 ^a^	5.2 ± 0.1 ^b^	6.6 ± 0.2 ^a^	7.3 ± 0.1 ^c^	<0.001
Colostrum intake (CI, g) **	534 ± 15 ^a^	458 ± 13 ^b^	574 ± 14 ^a^	632 ± 13 ^c^	<0.001
Relative CI (CI/kg BW) **	398 ± 10	408 ± 8	411 ± 9	384 ± 8	0.113
Colostrum intake CV (%) **	28.7 ± 1.4 ^a^	19.2 ± 1.2 ^b^	18.8 ± 1.3 ^b^	17.6 ± 1.2 ^b^	<0.001

* estimated with Devillers’s [26] equation; ** estimated with Theil’s [27] equation; ^a,b,c^ values in the same row with different superscript letters are significantly different.

**Table 3 animals-13-03100-t003:** Comparison of resident and cross-fostered piglets’ initial and final characteristics and performance.

		BW0 (kg)	W 24 h (kg)	WG0-24 (g)	CID (g)	CIT (g)	W 21 d (kg)	MR0-21 (%)
HET	RP	1.29 ± 0.03 (*n* = 130)	1.46 ± 0.03 (*n* = 127)	165 ± 8 (*n* = 127)	384 ± 13 (*n* = 127)	526 ± 16 (*n* = 127)	6.26 ± 0.14 (*n* = 114)	48.5
	CFP	1.38 ± 0.03 (*n* = 110)	1.55 ± 0.04 (*n* = 107)	165 ± 8 (*n* = 107)	378 ± 14 (*n* = 107)	532 ± 17 (*n* = 107)	6.65 ± 0.16 (*n* = 93)	51.5
	*p*-value	0.058	0.078	0.998	0.766	0.801	0.060	0.999
ULL	RP	1.01 ± 0.01 (*n* = 134)	1.23 ± 0.02 (*n* = 127)	128 ± 6 (*n* = 127)	315 ± 10 (*n* = 127)	429 ± 11 (*n* = 127)	5.38 ± 0.11 (*n* = 122)	36.7
	CFP	1.13 ± 0.01 (*n* = 190)	1.27 ± 0.01 (*n* = 183)	145 ± 5 (*n* = 183)	348 ± 9 (*n* = 183)	466 ± 9 (*n* = 183)	5.71 ± 0.10 (*n* = 171)	63.3
	*p*-value	0.152	0.074	0.026	0.013	0.012	0.027	0.071
	RP	1.39 ± 0.01 (*n* = 126)	1.58 ± 0.02 (*n* = 121)	186 ± 8 (*n* = 121)	418 ± 11 (*n* = 121)	573 ± 11 (*n* = 121)	6.39 ± 0.11 (*n* = 116)	45.0
UAL	CFP	1.40 ± 0.01 (*n* = 150)	1.58 ± 0.02 (*n* = 144)	183 ± 7 (*n* = 144)	414 ± 10 (*n* = 144)	570 ± 10 (*n* = 144)	6.12 ± 0.10 (*n* = 139)	55.0
	*p*-value	0.833	0.918	0.780	0.785	0.803	0.078	0.752
	RP	1.67 ± 0.01 (*n* = 198)	1.86 ± 0.02 (*n* = 192)	190 ± 7 (*n* = 192)	439 ± 13 (*n* = 192)	633 ± 14 (*n* = 192)	6.50 ± 0.10 (*n* = 116)	53.6
UHL	CFP	1.63 ± 0.02 (*n* = 138)	1.82 ± 0.02 (*n* = 133)	192 ± 9 (*n* = 133)	422 ± 16 (*n* = 133)	610 ± 16 (*n* = 133)	6.91 ± 0.12 (*n* = 139)	46.4
	*p*-value	0.080	0.127	0.833	0.422	0.297	0.009	0.789

BW—weight at time 0; W 24 h—weight at 24 h; WG0-24—weight gain from 0 to 24 h; CID—colostrum intake estimated with Devillers’s equation [26]; CIT—colostrum intake estimated with Theil’s equation [27]; W 21 d—weight at 21 days of age; MR0-21—mortality rate from time 0 until day 21 (% of each piglet type in the litter-type mortality); HET—heterogenous litters; ULL—uniform light litters; UAL—uniform average litters; UHL—uniform heavy litters; RP—resident piglet (nursed by the natural mother); CFP—cross-fostered piglet (nursed by other sow).

**Table 4 animals-13-03100-t004:** Litters and piglets’ characteristics at 21 days of age.

	Heterogenous (HET)	Uniform Light (ULL)	Uniform Average (UAL)	Uniform Heavy (UHL)	*p*-Value
*n*	20	27	23	28	
Litter size at 21 d	10.5 ± 0.2	10.9 ± 0.2	11.1 ± 0.2	11.0 ± 0.2	0.106
Litter weight at 21 d (kg)	66.6 ± 2.3 ^ab^	60.5 ± 2.0 ^a^	69.2 ± 2.1 ^b^	73.3 ± 1.9 ^b^	<0.001
Mean piglet weight 21 d (g)	6.4 ± 0.2 ^a^	5.6 ± 0.1 ^b^	6.2 ± 0.2 ^a^	6.7 ± 0.1 ^a^	<0.001
Within-litter CV at 21 d (%)	21.1 ± 1.5	17.5 ± 1.3	17.3 ± 1.4	16.6 ± 1.3	0.121
Litter weight gain 0–21 d (kg)	50.6 ± 2.2	47.1 ± 1.9	52.4 ± 2.1	53.5 ± 1.9	0.097
Mortality rate 0–21 d (%)	13.8	9.6	7.6	8.3	-

^a,b^ Values in the same row with different superscript letters are significantly different.

## Data Availability

Data will not be shared, due to privacy issues.

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
