# Peer review of "Effects of the Mean Weight of Uniform Litters on Sows and Offspring Performance"

_animals, 2023, doi:10.3390/ani13193100_

Round 1
Reviewer 1 Report
Good afternoon!
Thanks to the authors for their work, however, I would like to clarify some issues.
1. Did you check the effect of seasons on the heterogeneity of pig litter?
2. Why did you not perform necropsy of piglets?
3. Is it appropriate to use the Kruskal-Wallis test if it is used when comparing more than 2 groups and you only analyze two groups (cross-fostered piglets and resident piglets)?
4. What are the figures in Table 1?
5. Please bring Table 3 to a homogeneous format.
6. Has the quality of colostrum been checked in pigs?
7. In your earlier studies you described similar findings. How do past articles differ meaningfully from others you have previously written?
8. The conclusions of your paper are not clear, do you suggest weaning piglets at birth and forming homogeneous litters by weight?
Thanks for your work and good luck!
Reviewer 2 Report
Piglet survival and performance until weaning plays a key role in the sustainable development of pig farms. In this paper, the authors attempted to demonstrate the effects of within-litter weight variability and different average litter weights on the colostrum milk yield of sows and the growth performance of piglets. The structure of the article is clear and logical, but there are also some defects as follows.
1. The 98 experimental sows were from 12 farrowing batches. Were the sows in each farrowing batch fed the same diet? At the same time, it is recommended that the diet formula table be included in the article, as nutrition is also important for the colostrum composition of sows.
2. Why are the CV values in rows 115 and 116 different from the Intra-litter CV column in Table 1?
3. For the convenience of readers, please list the prediction equations of CY and CI directly under Table 2.
4. It is recommended that Tables 2 and 3 be placed next to each other.
5. For the convenience of readers, in the results section, when analyzing the content of Table 3, please mark it as from Table 3.
6. Please use the same font size as in Table 3, such as "(130)" and "(127)", and the words or numbers do not need to be bold.
7. It is suggested that the content of 3.2.2 should be divided into two subheadings for easy reading, such as the performance of piglets from 0 to 24 h and the performance of piglets at 0-21 d.
8. It is recommended that mortality rates from 0 -21d be added to Table 4.
English language fine. No issues detected
Author Response
Please see the attachement.

Reviewer 3 Report
The research may be interesting to readers, but formatting errors should be corrected.
Remember that the simple summary and the abstract should not exceed 200 words, check the journal format.
Homogenize the use of p or p in statistical differences throughout the document.
Table 1. Improve the table since it shows a moved format.
Table 4.- add at the bottom of the table the meaning of the literals a,b.
Check that the format of the citations is correct.
Author Response
Please see attachement.
